# Slovenia’s Food-Based Dietary Guidelines 2024: Eating for Health and the Planet

**DOI:** 10.3390/foods13193026

**Published:** 2024-09-24

**Authors:** Zlatko Fras, Borut Jug, Boštjan Jakše, Samo Kreft, Nina Mikec, Žiga Malek, Martina Bavec, Ana Vovk, Ana Frelih-Larsen, Nataša Fidler Mis

**Affiliations:** 1Faculty of Medicine, University of Ljubljana, 1000 Ljubljana, Slovenia; zlatko.fras@kclj.si (Z.F.); borut.jug@kclj.si (B.J.); 2Division of Medicine, Centre for Preventive Cardiology, University Medical Centre, 1000 Ljubljana, Slovenia; 3Department of Vascular Disease, University Medical Center Ljubljana, 1000 Ljubljana, Slovenia; 4Independent Researcher, 4280 Kranjska Gora, Slovenia; 5Faculty of Pharmacy, University of Ljubljana, 1000 Ljubljana, Slovenia; samo.kreft@ffa.uni-lj.si; 6Department of Molecular and Biomedical Sciences, Jožef Stefan Institute, 1000 Ljubljana, Slovenia; nina.mikec@ijs.si; 7International Institute for Applied Systems Analysis (IIASA), A-2361 Laxenburg, Austria; malekz@iiasa.ac.at; 8Biotechnical Faculty, University of Ljubljana, 1000 Ljubljana, Slovenia; 9Faculty of Agriculture and Life Sciences, Institute for Organic Farming, University of Maribor, 2211 Hoče, Slovenia; martina.bavec@um.si; 10Faculty of Arts, Department of Geography, University of Maribor, 2000 Maribor, Slovenia; ana.vovk@um.si; 11Ecologic Institute, 10717 Berlin, Germany; ana.frelih-larsen@ecologic.eu; 12Ministry of Health, 1000 Ljubljana, Slovenia; natasa.fidler@gmail.com

**Keywords:** dietary guidelines, food-based, nutrition, diet, environment, food system, Slovenia, food guide pyramid

## Abstract

The dietary guidelines of Slovenia, ‘12 Steps to Healthy Eating’, were first published in 2000 and revised in 2011. The ‘Food Guide Pyramid’ was initially published in 2000 and subsequently revised in 2015. ‘The Healthy Plate’ was first introduced in 2007. In February 2023, the Slovenian Strategic Council for Nutrition proposed new Food-Based Dietary Guidelines (FBDGs) that integrate both health and environmental considerations. In September 2023, the creation of new FBDGs was included in the Action Plan for implementing the Resolution on the National Program on Nutrition and Physical Activity for Health 2015–2025. In October 2023, the Ministry of Health of Slovenia appointed the core working group of 10 multidisciplinary experts from fields such as nutrition, food science medicine, public health, environment, pharmacy, and agriculture led by Prof. Dr. Nataša Fidler Mis, who drafted the guidelines. In February 2024, the World Health Organization Regional Office for Europe organized a virtual international workshop to assist Slovenia in developing food-based dietary guidelines. In May 2024, an international expert meeting was organized by the Ministry of Health of Slovenia, the Ministry of the Environment, Climate, and Energy, and the National Institute of Public Health of Slovenia to present the first scientific draft of the SLO FBDG for external international peer review. The meeting included lectures from world-leading experts to present healthy diets from sustainable food systems, integrate climate and sustainability aspects into the new SLO FBDG, discuss the findings with the Slovenian core working group, extended working group of the SLO FBDG, and invited experts. The final version of SLO FBDG is expected to be released by the end of 2024.

## 1. Introduction

The growing body of new scientific research on nutrition and its impact on health and disease requires the state to analyze and potentially incorporate these findings into updated dietary guidelines. These guidelines should address the most urgent public health issues (common chronic noncommunicable diseases), environmental concerns, and the food system or national food sovereignty [1].

The Slovenian dietary guidelines, known as the 12 Steps to Healthy Eating, were originally released in 2000 and were updated in 2011. Similarly, the ‘Food Guide Pyramid’ was first published in 2000 and then revised in 2015. In 2007, ‘The Healthy Plate’ was introduced. These updated visual aids, presented on a single page, do not include references and are intended to assist individuals in following the recommended diet [2]. The aim of this study is to present the procedure for developing new Slovenian Food-Based Dietary Guidelines (SLO FBDG), especially the initial two phases: (i) initial work of the core working group and (ii) international expert meetings.

The Slovenian Strategic Council for Nutrition proposed a new FBDG that integrates both health and environmental considerations in February 2023 [3]. The Action Plan for implementing the Resolution on the National Program on Nutrition and Physical Activity for Health 2015–2025 included developing a new SLO FBDG in September 2023 [4,5]. In October 2023, the Ministry of Health of Slovenia appointed a core working group of experts to write the guidelines. The multidisciplinary core working group of experts began drafting the guidelines. The core working group consisted of ten nutritional and sustainability experts from various fields, including nutrition, medicine, public health, the environment, pharmacy, and agriculture. The development of the SLO FBDG is funded by the Ministry of the Environment, Climate, and Energy and the Ministry of Health. In May 2024, the minister of health of Slovenia appointed an extended working group of 33 multidisciplinary experts from Slovenia and six international reviewers to write feedback and review the guidelines on a specially prepared online formula with precise instructions on how to prepare feedback and review. The transparent and democratic process of FBDG development, which uses high scientific standards (systematic reviews/meta-analysis), includes six phases: (i) initial work of the core working group; (ii) international expert meetings (in February 2024 and in May 2024 (this report)); (iii) extended working group and external international peer review; (iv) public consultation; (v) final version (professional part and for the public); (vi) approval; and (VII) dissemination, implementation, and evaluation. In February 2024, the World Health Organization (WHO) Regional Office for Europe collaborated with the WHO Country Office in Slovenia to organize a virtual international workshop to support Slovenia in developing FBDG. This initiative involved the participation of global nutrition, environmental, and public health experts, including Prof. Dr. Walter Willett and Prof. Joseph Poore.

The initial work of the core working group encompassed a systematic literature search and writing of the first scientific draft of the SLO FBDG. They provide evidence-based recommendations for promoting foods and nutrients associated with improved health outcomes and sustainable environmental impacts. The guidelines were developed under the auspices of the Ministry of Health by a multidisciplinary expert group through peer review and public and professional discussion. The guidelines were developed in response to appraisals of current dietary patterns and the prevalence of noncommunicable diseases (NCDs) in the adult Slovenian population concerning the food system, eating culture, and lifestyle. In Slovenia, the Mediterranean diet, which promotes the consumption of more vegetables and fruits and, at the same time, less saturated fatty acids from animal sources as well as processed foods, has been recommended for 20 years [6,7]. The transition towards a predominantly plant-based diet is supported by recent scientific evidence highlighting the health benefits of a plant-based diet and its positive environmental impacts. Similar steps toward plant-based dietary recommendations have been adopted worldwide, as evidenced by the EAT/Lancet Commission [8], the Canadian dietary guidelines [9], the Nordic Nutrition Recommendations (NNR) 2023 [10], the Danish FBDG [11], and the recent German FBDG [12].

The recommendations provide a wide range of dietary options that can be customized to suit each person’s unique needs or preferences. By tailoring food groups to individual requirements, we can promote sustainable and healthy eating habits among adults of all ages. In addition, the guidelines provide valuable information on essential nutrients and, more significantly, will also encompass recommendations for meal planning and food preparation. The guidelines also encourage regular physical activity as an essential major determinant of a healthy lifestyle, being part of the comprehensive national strategy to promote health and prevent NCDs to the highest possible level for almost two decades [6,13,14].

In summary, the SLO FBDG recommends a predominantly plant-based diet (high intake of vegetables, fruits, whole grains, pulses, potatoes, nuts, and seeds) with moderate intakes of dairy products, eggs, and fish; limited intake of meat; and minimal (ideally no) intakes of processed meat, alcohol, and processed foods high in saturated fats, salt, refined starch/grains, and added/free sugar. Drinking plain water, mineral water, or nonsweetened tea is recommended as the primary beverage.

In May 2024, the Ministry of Health of Slovenia, the Ministry of the Environment, Climate, and Energy, and the National Institute of Public Health of Slovenia organized an international expert meeting in Ljubljana, Slovenia. The expert meeting aimed to present the first scientific draft of the SLO FBDG to 33 extended working group members, five invited speakers, and six reviewers. World-leading experts were invited to present healthy diets from sustainable food systems, integrate climate and sustainability into the new SLO FBDG, and discuss the findings with the Slovenian core and extended working group.

## 2. May Expert Meeting Main Objectives

(a)Invited world-leading experts to present healthy diets from sustainable food systems.(b)SLO FBDG core working group members presented the following topics: (a) the current draft of SLO FBDG, which will be the main foundation for food and health policies in Slovenia; (b) the project (2023–2024) involving nutrition, diet, health, sustainability, and methodology scientists. The new SLO FBDG will, for the first time, integrate climate and other sustainability aspects, with the final version expected at the end of 2024.(c)Invited experts from Slovenia had a unique opportunity to discuss SG2024 with international, world-leading experts and SLO FBDG core working group members. The event featured five lectures from prominent international experts: Dr. Eric Feigl-Ding from the U.S., Prof. Paul Behrens from the Netherlands, Prof. Marco Springmann from the UK, Anne Carolin Schäfer, M.Sc. from Germany, and Iben Humble Kristensen, M.Sc. from Denmark. Additionally, the Slovenian working group members delivered ten lectures.

The five presentations by international experts were as follows:

Dr. Eric Feigl-Ding presented a lecture titled “EAT-Lancet Commission on Healthy Diets from Sustainable Food Systems”. This lecture was based on Willett et al.’s publication in the Lancet and focused on how the findings can be applied to update contemporary FBDGs, emphasizing a shift towards plant-based nutrition for both health benefits and sustainability [8].

Prof. Paul Behrens delivered a lecture on “The Role of Plant-Based Shifts in Reducing Environmental Impacts While Increasing Resilience and Food Security”. He discussed how transitioning to plant-based diets can mitigate environmental impacts while enhancing food security and resilience.

Prof. Marco Springmann presented “Reducing the Environmental and Health Impacts of Our Food System: Preliminary Analysis of new SLO FBDGs”. He discussed his analysis using a comparative risk framework on the burden of disease mortality linked to 10 risk factors (including high red and processed meat intake; low intake of fruits, vegetables, nuts, legumes, and whole grains; and issues related to underweight, overweight, and obesity) and 5 causes of death (congenital heart disease, stroke, type 2 diabetes mellitus, cancer, and respiratory diseases). His findings suggest that new SLO FBDGs could reduce the risk of death by 80% compared with the previous guidelines, potentially preventing 1500 deaths annually. Additionally, the new SLO FBDGs have a much lower environmental impact, aligning closely with planetary boundaries (approximately 100% of the limits), whereas the old guidelines exceeded these boundaries by 350%.

Anne Carolin Schäfer provided an “Overview of the New German FBDG”, detailing the process of forming the German Nutrition Society’s (DGE) working group, the methodology consultation in 2022/2023, and the final derivation of the FBDGs in 2023/2024. The new German FBDGs, developed via an optimization model, recommend a predominantly plant-based diet, with over 75% of foods being of plant-based origin.

Iben Humble Kristensen presented an “Overview of Food-Based Dietary Guidelines (FBDG) and the Danish FBDG”, discussing several new European and international guidelines that are increasingly shifting towards predominantly plant-based nutrition owing to health and environmental considerations.

The presentations by the Slovenian FBDG core working group members covered a wide range of topics related to the development and impact of the SLO FBDGs:

Prof. Nataša Fidler Mis presented “The Process of Developing the SLO FBDG”, outlining the methodology and steps involved in creating the guidelines.

Prof. Zlatko Fras delivered a lecture titled “SLO FBDG: Impact on Health and Sustainability”, discussing what the SLO FBDG entails, who can benefit from it, its role in addressing NCDs, and how it can optimize health and sustainability, including embracing the planetary diet.

Prof. Borut Jug, Prof. Samo Kreft, and Prof. Nataša Fidler Mis collaborated to present “Food Group Recommendations in the SLO FBDG”. They discussed recommendations for 16 food groups, including cereals, potatoes, and other starchy tubers; pulses/legumes; fruits, vegetables, nuts, and seeds; fish and seafood; milk and dairy products; meat and processed meat; eggs; fats and oils; herbs and spices; sweets and snacks; water and nonalcoholic beverages; and alcohol; and ultra-processed foods.

Dr. Boštjan Jakše presented the criteria for a scoring system model used to classify foods as ‘high in’ sodium, saturated fats, and fiber and as a ‘source of’ proteins, omega-3 polyunsaturated fatty acids (n-3 PUFAs), and micronutrients. This model is based on guidelines from the European Food Safety Authority (EFSA), the Netherlands Nutrition Center, and other European recommendations. It has been adapted for Slovenia using data from the Slovenian food database [15,16,17,18,19,20,21].

Dr. Boštjan Jakše also presented “Comments on Energy and Nutrient Intake in SLO FBDG”, addressing energy intake and the intake of macronutrients (protein, carbohydrates, dietary fiber, fats, and dietary cholesterol) and micronutrients (including vitamins B12, C, D, folate, iron, magnesium, potassium, calcium, sodium, zinc, iodine, and selenium).

Dr. Žiga Malek discussed “Reducing Environmental Impacts of Slovenian Diets”, focusing on the unsustainability of current food systems, the need for a planetary focus on Slovenian diets, the environmental impacts of food across its life cycle, and how consumers can reduce these impacts.

Prof. Zlatko Fras and Prof. Martina Bavec presented on “SLO FBDG: Shaping a Healthier Future: Health Promotion, Physical Activity, and Organic Foods”. Prof. Fras discussed the role of SLO FBDGs in promoting health. He used scientific studies to emphasize the importance of informing people about the benefits, risks, and challenges of practicing various dietary patterns. These patterns included the Mediterranean diet, DASH diet, vegetarian diets, low-carbohydrate, high-fat diets, and ABO blood group diet, as well as various forms of fasting. Professor Bavec covered the nutritional and environmental benefits of organic foods.

The members of the extended working group and external reviewers were asked to send their written comments on the draft version of the SLO FBDG via an online form within four weeks after the May 2024 meeting.

## 3. Conclusions and Future Directions

Food systems significantly impact sustainability at both the local and global levels. Slovenia aims to transform its food system by involving stakeholders, such as farmers, food processors, consumers, and policymakers, to create a healthy and sustainable system. We aimed to outline the development process of a new SLO FBDG, particularly the initial stages. However, it is important to note that this study does not include the results of public consultation; the final versions of the SLO FBDG (both professional and public versions); approval, dissemination, implementation, and evaluation.

After the May Expert meeting, we received feedback from international reviewers and Slovenian extended working group members of SLO FBDG within four weeks. This feedback will be used to create the second draft of SLO FBDG, covering the nutritional and environmental aspects. Following this, we (the core working group) will write the SLO FBDG for professionals and for the public that will be available for public consultation. We will include remarks in the final version and the final version of the document, which will be submitted to the national assembly for approval.

In summary, transitioning to a diet emphasizing plant-based foods and reducing the use of animal-based products provides substantial health and environmental sustainability benefits. This shift can decrease the risk of chronic diseases, improve overall well-being, and mitigate environmental damage. Furthermore, healthier eating habits can increase productivity and lower healthcare costs, positively impacting the Slovenian economy. However, achieving this transition requires more than just dietary guidelines; it demands education and supportive policies now and, in the future, to successfully shift towards a healthy and sustainable diet.

## Data Availability

No new data were created or analyzed in this study. Data sharing is not applicable to this article.

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
