# Peer review of "Slovenia’s Food-Based Dietary Guidelines 2024: Eating for Health and the Planet"

_foods, 2024, doi:10.3390/foods13193026_

Round 1
Reviewer 1 Report
Comments and Suggestions for Authors
Dear Authors,
The commentary on development of the process of FBDG n Slovenia is well and clearly written, providing sufficient information on the steps taken to develop national dietary guidelines with environmental consideration. Well done, no further comments.
Reviewer 2 Report
Comments and Suggestions for Authors
Dear Editor,
Although the article does not have the standard structure of a research, the topic is very important from the point of view of food, health, environmental issues and sustainability. Slovenia took the initiative to renew the FDBG considering the aspects that concern the world population and the papaer describes the steps involved in this process.
Reviewer 3 Report
Comments and Suggestions for Authors
Review for the manuscript
Slovenia’s Food-Based Dietary Guidelines 2024. Eating for Health and The Planet
Dear Editor,
Thank you for the invitation to review for FOODS. Please find below my comments and suggestions.
TITLE
It is adequate.
ABSTRACT
In this section, in lines 34-35 we find: " …and the National Institute of Public Health of Slovenia in Slovenia to present the 34 first scientific draft of the SLO FBDG for external international peer review".
I suggest: …and the National Institute of Public Health of Slovenia to present the 34 first scientific draft of the SLO FBDG for external international peer review
In lines 34-39 it is possible to read that “The meeting included invited lectures from world-leading experts to present healthy diets from sustainable food systems, integrate climate and sustainability aspects into the new SLO FBDG, and discussed the findings with the Slovenian core working group, extended working group of the SLO FBDG and invited 38 experts. The final version of SLO FBDG is expected to be released by the end of 2024.”
I also suggest including the results of the discussions that were held in this section. What actions were taken or planned based on these meetings?
KEYWORDS
The chosen keywords are Keywords: dietary guidelines; food-based; nutrition; diet; environ
ment; food system; Slovenia. I suggest including among them “Food Guide Pyramid”.
INTRODUCTION
In this section, I suggest to
1- Include more and new references. Please show how this manuscript is important and innovative based on the current publications that the authors can find in PUBEMD, Google Scholar, Embase, etc.
2- The aim of the study is not clear.
3- In topic 2. May Expert meeting main objectives:
I do not see the need to cite the conference titles and the names of the scientists who participated in the lectures. Is it necessary to include this topic?
METHODS
NA
CONCLUSIONS and FUTURE DIRECTIONS
These sections are important to be carefully addressed; please revise
However, I suggest including the limitations of this review. Please build a sentence deeply discussing the future perspectives for this project.
Please include the limitations of this study.
Comments on the Quality of English Language
Only minor.
Round 2
Reviewer 3 Report
Comments and Suggestions for Authors
Dear authors,
Thank you for taking my suggestions into account.
Dear Doctor,
I believe that the manuscript can be published.
With best regards,
Sandra M. Barbalho